# COVID-QA: A Question Answering Dataset for COVID-19

**Timo Möller**[*]
deepset GmbH
Berlin, Germany

**G Anthony Reina**[*]
Intel Corporation
Santa Clara, CA, USA

**Raghavan Jayakumar**
Lawrence Livermore
National Laboratory
(retired)
Livermore, CA, USA

**Malte Pietsch**
deepset GmbH
Berlin, Germany

## Abstract

We present COVID-QA, a Question Answering dataset consisting of 2,019 question/answer pairs annotated by volunteer biomedical experts on scientific articles related to COVID-19. To evaluate the dataset we compared a RoBERTa base model fine-tuned on SQuAD with the same model trained on SQuAD and our COVID-QA dataset. We found that the additional training on this domain-specific data leads to significant gains in performance. Both the trained model and the annotated dataset have been open-sourced at: https://github.com/deepset-ai/COVID-QA.

## 1 Summary

We selected 147 scientific articles mostly related to COVID-19 from the CORD-19 (The White House Office of Science and Technology Policy, 2020 (accessed May 9, 2020)) collection to be annotated by 15 experts. Although the annotators were volunteers, it was required that all have at least a Master's degree in biomedical sciences. The annotation team was led by a medical doctor (G.A.R.) who vetted the volunteer's credentials and manually verified each question/answer pair produced. We used an existing, web-based annotation tool that had been created by deepset[1] and is available at their Neural Search framework haystack[2]. The annotations were created in SQuAD (Rajpurkar et al., 2016) style fashion where annotators mark text as answers and formulate corresponding questions. COVID-QA differs from SQuAD in that answers come from longer texts (6118.5 vs 153.2 tokens), answers are generally longer (13.9 vs. 3.2 words) and it does not contain n-way annotated development nor test sets. We chose a RoBERTa-base architecture (Liu et al., 2019) and fine-tuned it on the

---

| Model | Exact Match | F1 |
|---|---|---|
| Baseline | 21.84 | 49.43 |
| COVID-QA model | 25.90 | 59.53 |

Table 1: Performance comparison. The scores for the COVID-QA model are averaged across the 5 folds.

SQuAD dataset as a baseline model. We continued training the baseline model using our COVID-QA annotations in 5-fold cross validation manner.

Table 1 shows the performance of the baseline model vs. the model finetuned on COVID-QA. Finetuning the model on COVID-QA results in significant improvement across both metrics though the overall scores are pretty low compared to SQuAD. We hypothesize the low scores relate to more complex question/answer pairs on much longer documents and the lack of multiple annotations per question. Without n-way annotations it is difficult to assess the human benchmark and thereby the true difficulty of the dataset. We might add n-way annotations in a future version of the dataset.

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
