# OpenReview forum: "COVID-QA: A Question Answering Dataset for COVID-19"
_aclweb.org/ACL/2020/Workshop/NLP-COVID — NLP-COVID-2020 Abstractonly_

### Official Review · AnonReviewer1 · 2020-06-19
**Development of a QA dataset for COVID-19; a work in progress that requires revision**

**Rating:** 5
**Confidence:** 4

**Review:**

Authors have developed a QA dataset based on scientific articles about coronaviruses released as part of the COVID-19 Open Research Dataset (CORD-19). A small set of 450 articles were selected based on keyword matching, random sampling, and expert search, and around 2K Q-A pairs were developed against these articles, where question were generated by annotators and answers were sentences in the articles. Annotators were required to have a degree in the biomedical sciences. Also, the annotated dataset and a fine-tuned BERT-based model have been released.

Authors investigate an important problem, and the paper is well organized and well written. In addition, such a dataset could be a valuable resource as it enables computer scientists and clinicians to address their information need about COVID-19 based on reliable sources (scientific literature).

However, there are few shortcomings that prevents me from recommending the paper for acceptance at this time:

1. Data Quality: Each QA pair has been annotated by only one annotator. I suggest having the data annotated by at least two annotators followed by a third one (or a discussion between the first two annotators) to adjudicate the disagreements. It's important to consider that a good model should learn the actual task as apposed to optimally mimicking the behavior of a single annotator. In addition, I wonder if annotators labeled "all" sentences of articles that answer the questions. This is important for correct evaluation (models should not be penalized for correct predictions that are not labeled). In addition, I found many questions in the dataset that:

a): are not necessarily relevant COVID-19:
- "What is the main cause of HIV-1 infection in children?"
- "What plays the crucial role in the Mother to Child Transmission of HIV-1 and what increases the risk"
- "What is the molecular structure of bovine coronavirus?"
- "What causes the outbreak of SARS and MERS?"


b) are general with potentially multiple correct answers:
- "Who plays a role in regulating the immune response to a vital infection?"
- "Which species are more prevalent but less severe?"
- "What is a significant cause of Influenze like illness among healthy adolescents and adults presenting for medical evaluation?"


c) combined several answers together:
- Q: "Which species are more prevalent but less severe?"
  A: "HCoV-HKU1, HCoV-OC43, HCoV-NL63, and HCoV-229E"


In addition, "no answer" questions are probably the most interesting and prevalent questions about COVID-19. Such questions have not been developed in the context of this work.


2. Modeling: Articles were splitted into multiple chunks of text and only two chucks (a positive chunk containing the answer of a given question and a randomly sampled chunk as negative example) were used to test the model. This design choice can oversimplify the task. In addition, given that the "no answer" output of the model was disabled, I'm not sure how the model works on the negative chunks. What should the model predict in cases of negative chunks?


3. Evaluation: Top-n-accuracy measure is defined as follows: "it compares the gold label against n model predictions and looks for any overlap between prediction and answer positions." It is not clear how accuracy is derived from overlaps. But, if "any" overlap between prediction and answer positions is treated as a hit, then Top-n-accuracy is not a good/reliable metric because a small overlap (e.g. a stopword) is just enough to treat a wrong prediction as a correct answer, and an untrustworthy algorithm as a trustworthy one. This is specially problematic for questions with long answers.

I recommend authors to consider and address the above comments and re-submit their papers.

---

> ### Author Response · Authors · 2020-06-23
> **Responses to all points mentioned - an ordered list based on our perceived importance**
>
> We highly appreciate the timely as well as accurate review of our efforts and are motivated to improve upon our work with the help of this review. The mentioned points are all relevant and show a high degree of insights into the domain and a thorough analysis of the paper and especially the accompanying annotated dataset. We hope to address each issue in a satisfactory manner. We will especially address changes to the paper - significant changes to the dataset are, by nature of a volunteer led expert annotation project, out of scope before submission deadline. Thanks also for acknowledging strong points of our paper, that an important problem is investigated and the paper is well organized as well as well written.
>
> We reordered your points based on our perception of how important they are to address.
>
> >1. Data Quality: Each QA pair has been annotated by only one annotator.
>
> Agreed. Multiple annotations per QA pair are very important. The current version of the dataset misses these n-way annotations, a limitation which we state in "5.3 Dataset Improvements".
> We manually checked QA pairs by experts that were familiar with both the label tool and the domain. These manual checks are unfortunately left undocumented.
> We also set up a slack workspace for volunteers to discuss certain annotations. We might be able to address this in a future version of the dataset.
> We added details of the manual checking under “3.1 The Annotation Process”
>
> > 2. Modeling: unfair test scenario
>
> There might be a slight misunderstanding. Only training data is downsampled, tests are done on the whole article.
> This downsampling made training much faster since there are on average ~36 negative passages per each positive passage.
> Testing is done on the whole article, so we believe the test scenario is not unfair.
> We made the downsampling procedure more explicit under “4.3  Evaluation” and also emphasize the size of the documents (6118.5 mean tokens per doc COVID-QA vs 153.2 tokens for SQuAD) as being a challenging test scenario under “5.1  Interpretation of Results”.
>
> > 3. Evaluation: Top-n-accuracy
>
> You are correct, "any" overlap creates a binary hit or miss scenario so accuracy can be computed. It is important to note that overlap in position is used, not overlap of strings.
> We hope the positional overlap on long documents reduces the reliability concern. To be more certain we analyzed a subset of predictions, available at https://github.com/deepset-ai/COVID-QA/blob/master/data/question-answering/predictions.tsv, and cannot find meaningless predictions (e.g. stopwords).
> We made the paper more explicit about how accuracy is calculated under “4.2  Metric”.
>
> > 1. Data Quality: a): are not necessarily relevant COVID-19
>
> The articles were from the CORD-19 dataset which originally contained a fairly generic set of scientific articles. The number of COVID-19-specific ones is understandably low given the newness of the outbreak. We attempted to narrow the CORD-19 articles even further by extracting more relevant subsets. Nevertheless, SARS and MERS have similar domain-specific relevance to COVID-19 since they define similar viral pandemics. HIV-1 infection mechanisms (and indeed all viral infection mechanisms) are highlight relevant to COVID-19 researchers as well. In order to keep the COVID-QA model generalizable, we felt that including these non-COVID (but viral outbreak related) articles was highly relevant. We added a more detailed explanation of chosen articles under "3.2 Annotation Process" (first paragraph)
>
> > 1. Data Quality: "no answer" questions are probably the most interesting
>
> Agreed, "no answer" annotations would be very helpful in creating good QA systems and have not been included in this dataset as mentioned in the paper under "5.3 Dataset Improvements".
> Unfortunately we see true "no answer" annotations as out of scope for the current version for the following reasons:
> 1. True "no answers" as present in Natural Questions are hard to get, because we would first need
> naturally occuring questions - not questions constructed by looking at some text but questions by actual users using a search engine.
> Since we do not operate such a search engine we only might be able to address "no answers" in another version of the dataset, if we get search queries by one of the CORD-19 search engines.
> 2. One could artificially construct "no answers" as in SQuAD v2.0. The usefulness of those constructed "no answers" is debatable.
>
> > 1. Data Quality: b) are general with potentially multiple correct answers
>
> We added the internal discussions about multiple annotations per question to the paper under “3.1  The Annotation Process”.
>
> > 1. Data Quality: c) combined several answers together
>
> The combination of several answers only happens when the answer is consecutive text. If it were not consecutive text, this problem boils down to 1b above. Variance in answer length is a reason we stated for choosing top-n-accuracy in "4.2 Metric".

---

### Official Review · AnonReviewer2 · 2020-06-21
**QA dataset on CORD-19**

**Rating:** 6
**Confidence:** 4

**Review:**

Summary of Work: This work provides a new resource, a QA dataset built on top of CORD-19 annotated for span prediction task in form of SQuaD dataset. The dataset contains 2K question from 147 articles (I am not clear on this point, the authors say they extracted 445 documents). They also built a model on top of this dataset by finetuning roberta base and find improvement over original roberta base.

While the dataset may be useful, we don't really know what criteria the annotators used to annotate the questions (How do they select what to annotate as answer? , what is considered important? - Do note that the authors also provide online videos of instructions that I have note reviewed which may contain the answer to this.)

Also the authors mention the questions can be reviewed/corrected by other annotators but again I don't know if it was done for every question and by how many other annotators. The annotators are said to be volunteers (I assume that to mean they were not paid) but I am not sure how they were recruited and how their qualification were evaluated (the paper says the authors require them to have a masters in biomedical sciences, although not sure how that is checked).

In general, the paper provides us some info about the new dataset, but the details are left out. As such, I am not confident in its utility for useful (and trustable) research .

The modelling framework is straightforward although their may be obvious improvements that can be made by training on biomed corpora. The top-n-accuracy metric needs more explanation (does the overlap mean overlap in text or position -- the first one is useless since we overlap on stopword easily, the second one might be more useful, but more analysis is needed).

This work needs a spell check (caronavirus -> coronavirus, fee -> see, and more).

I would note that a lot of my criticism maybe resolved by providing more details. But without them, I cannot truly provide a good review of the dataset itself. Therefore, currently I am giving a marginally below threshold score (and not a clear rejection since the dataset *may* be useful)

UPDATE: The authors answered questions presented above. The concern still remains about the quality of annotations, as well as the utility of model released. On the basis of this, I have increased my score.

---

> ### Author Response · Authors · 2020-06-23
> **Responses to all points in order of occurrence**
>
> Thank you for this timely and detailed analysis of our paper. We are keen on adding more details to the paper. We left out some information because we did not want to overburden the reader in the first place but are happy to elaborate more.
> Following is a list of responses - we also mention where we applied the relevant changes to the paper.
> --------------
>
> > Clarification on number of articles annotated
>
> Good point, it is a bit unclear. We uploaded 450 articles to the annotation tool, 147 of those contain actual labels. We clarified this in the newer revision by stating: "Of these 450 scientific articles, 147 were annotated over the 8 week period [...]" under "3.1  The Annotation Process".
>
> > Annotation criteria
>
> You are right, the guidelines can be found inside the videos mentioned in the paper. For convenience (not everyone likes videos) we also uploaded the annotation guidelines as pdf and will linked them in "3.1  The Annotation Process". You can also find them here: https://github.com/deepset-ai/COVID-QA/blob/master/data/question-answering/Handbook%20-%20Labelling%20Tool.pdf
>
> > Correction of Annotations
>
> This is an important point, also made by the reviewer no. 1. Unfortunately we cannot give a fully satisfying answer. We manually checked some of the labels and found them to be of good quality as well as created a slack workspace where people could discuss specific annotations. We added information about manual checks under “3.1  The Annotation Process”.
> We hope to create a more thorough and especially documented review of quality checks in a future version of this dataset.
>
> > Recruiting of Annotators
>
> We have added how volunteering annotators were recruited to the paper under “3.1  The Annotation Process”.
>
> > Modelling Framework
>
> Agreed, the modelling is not the best you can get with current methods and we talk about improvements to the modelling part in "5.2 Modelling improvements" where we also state your suggestion of training on other/biomed data.
>
> > More Explanations for top-n-accuracy metric
>
> The overlap is on position as stated in the paper under "4.2 Metric": "[...] we choose to take multiple
> model predictions and any positional overlap between prediction and label as metric and refer to it
> as top-n-accuracy. Top-n-accuracy compares the
> gold label against n model predictions and looks for
> any overlap between prediction and answer positions. [...]"
> We made this explanation more prominent and more explicit in the new revision.
>
> > Spell Check
>
> Fixed, thanks for pointing out!

---

### Official Review · AnonReviewer4 · 2020-06-28
**Potentially a good resource**

**Rating:** 5
**Confidence:** 4

**Review:**

This work presents a question-answering dataset created on a subset of CORD-19. Obviously, having a large QA dataset is valuable and if done well, can be a useful research both in the bioNLP field. The dataset is created by finding "answers" to potential questions that the text under examination would answer, surely, this effort has taken many hours to create. For what has been done, much less time and effort is put into writing about it. The manuscript is vert fragmented and hard to follow:
- Abstract: Not clear what exactly is done and what contributions are
- Introduction: You could reduce explanation on CORD-19 (removing the name of organisations for example) and refer to it in a citation. Instead it was expected to see what the NLP problem here was and what are the contributions of this work. None of that is there.
- Related work is hardly a literature review. What is the common practice in the NLP community in creating a QA dataset. This is an old area. Surely there would be reputable (not arxiv) papers there to check and compare? Why tools are there and why are they relevant given they are two IR tools mentioned?
- Section 3 needs a full rewrite. All the important details are missing. What were the annotation instructions and why? Ho w many annotators did you have? Was there any quality control? Any inter-annotator agreements for example?
- Not sure if the  paragraph before 3.1 makes any sense. BioBERT and SciBERT and so on are more representations. What is the relevance here?
- In Section 3, it says annotators are told they are developing a "method". How is this a method?
- Method section is quite weak but maybe enough for a dataset work.
- Please check your references, even old ones are arxiv. Maybe they are published somewhere? or maybe you can find a peer-reviewed citation for some of those.

Overall, even with the urgency of publishing COVID-19 related work, the paper should have acceptable quality which I believe this one need revision to get there.

---

> ### Author Response · Authors · 2020-06-29
> **Mixed review giving some good suggestions for improvement but lacks foundation in other parts**
>
> > Reviewers Intro
>
> The reviewer might be lacking important knowledge of the domain because the dataset is not created by “finding answers to potential questions” as stated by the reviewer but instead “The researcher uses the mouse to highlight the answer (pink) and creates a question based on that answer (box on left).” (caption of Figure 1 in paper). This is also the annotation procedure of the prominent SQuAD dataset that the reviewer should be aware of.
>
> > Abstract: Not clear what exactly is done and what contributions are
>
> Please be more specific, because we believe the abstract is rather clear (“Our dataset consists of 2,019 question/answer pairs annotated by volunteer biomedical experts”) as well as states our contributions (dataset creation + “We found that the additional training on this domain-specific data leads to significant gains in performance”).
>
> > Introduction
>
> Agreed. The first part could be a bit more crisp while the latter part could be more elaborate. We left the first part untouched but added more details to the contribution as well as motivation.
>
> > Related work [usage of arxiv references and why are two IR tools mentioned?]
>
> - Thanks for the hint. We changed references for SQuAD and MRQA to their respective non-arxiv versions. Other references in the related work are, given the recent nature of the crisis, preprints.
> - The last sentence motivates the mention of IR tools: “We view our work as a complementary effort to aid in the evaluation and/or improvement to these existing tools”. Maybe you are aware of the discussion around the submission “Rapidly Deploying a Neural Search Engine for the COVID-19 Open Research Dataset” where a lot of criticism revolves around the lack of proper evaluation. With our QA dataset these existing tools could be potentially evaluated.
>
> > Section 3 needs a full rewrite. All the important details are missing. What were the annotation instructions and why? How many annotators did you have? Was there any quality control? Any inter-annotator agreements for example?
>
> - Possibly the reviewer failed to notice the hyperlinks to detailed annotation guidelines as well as the videos explaining the annotation procedure. Possibly the reviewer printed out the paper? We had an internal discussion about footnotes vs hyperlinks and thought hyperlinks would be more useful in a paper linking many external resources + people cannot follow or click on footnotes printed on paper anyways
> - General remark. During the review process we added a lot of additional material especially to section 3. Looking now at the section we realized we should divide it into smaller and more digestible parts, which we did by introducing subsections “3.1  Article and Annotator Selection”, “3.2  QA Annotation Process” and  “3.3  Dataset Analysis”
> - Number of annotators: Very good point. We added it to section "3.1  Article and Annotator Selection"
> - Was there quality control: Quality control is stated in the paper that each annotation was verified by the annotation team lead. We made this more explicit in the newest revision in section "3.2  QA Annotation Process", first passage.
> - Inter annotator agreement (also criticized by other reviewers): as stated in the discussions part the dataset does not contain multiple annotations per question. So inter annotator agreement is not possible to compute. We state the reasons for this in the discussion section as well.
>
> > Section 3. What is the relevance of BioBERT and SciBERT
>
> True, Bio- and SciBert are Language Models for creating representations (of domain text), these domain representations can be used for doing domain specific QA. All these domain efforts have in common to improve the performance in that domain. This was our hope when creating COVID-QA, improving performance of search systems in this domain.
>
> > Section 3, it says annotators are told they are developing a "method". How is this a method?
>
> We agree, this is a confusing choice of words and change “method” to “datapoints”.
>
> > Method section is quite weak but maybe enough for a dataset work.
>
> Thanks, we take this as a compliment.
>
> > Please check your references
>
> Agreed, referencing preprints instead of peer reviewed publications is not good practice. We went through the references and changed them where applicable.

---

> > ### Comment · AnonReviewer4 · 2020-07-02
> > **Answer to the authors**
> >
> > Thanks for the clarifications.
> >
> > I just want to add that my concerns were to help improve the writing of the paper. When you have a dataset paper, referring your reader to hyperlinks is a bit of a stretch and some of the annotation guidelines and important annotation related information must stay in the paper. I had looked in those links but I'm not reviewing those but what you have written here.
> >
> > For the current draft:
> > - The abstract still needs improvement. Here's what the flow breaks: "...This presents, to
> > date, the largest manually created Question Answering dataset on Covid-19 related material. We began with a RoBERTa base model that was initially" Abruptly RoBERTa is mentioned. Again this is not a method paper so you could simply say we used benchmarked our dataset using X methods.
> >
> >
> > - Tools: Again, I'd think there are tools that could be more QA and less IR? Or are there none? No objection to keep IR tools but there should be something on QA as it's most relevant to you.

---

> > > ### Author Response · Authors · 2020-07-02
> > > **Response**
> > >
> > > Thanks for the prompt answer. We really appreciate these more specific suggestions because they outline potential for improvement.
> > >
> > >
> > > > annotation guidelines and important annotation related information must stay in the paper
> > >
> > > We agree that some of the annotation guidelines need to be in the paper. Though we largely followed "the SQuAD style label process" which is well known and we reference in the paper. We included where we diverged from the SQuAD dataset creation process: annotating on longer documents, missing n-way annotations and possibly less controlled quality checks. We believe omitting details that referenced papers explain is a reasonable thing to do but are open for discussion.
> > >
> > > > The abstract still needs improvement. Here's what the flow breaks [...]
> > >
> > > Thanks for specification. We acknowledge that it is a break of flow, but would argue that this break is common practice: creating a dataset, then evaluating this dataset. For more clarity we could add in another revision "To evaluate the dataset we began with ..."
> > > Concerning the use of RoBERTa: We feel that BERT or RoBERTa are nowadays well known for people to understand the relation between a QA dataset and a RoBERTa model. Do you have a proposal to state the evaluation more clearly while being concise and to the point?
> > >
> > >
> > > > Tools: Again, I'd think there are tools that could be more QA and less IR?
> > >
> > > It is difficult to apply pure BERT QA models (joint computation of question and document in one model) in real time to incoming user questions on a whole database of documents. So realistic user facing tools should use IR related components. Nevertheless the tools stated do not just retrieve documents but highlight potential answers, which we believe is more QA than IR.

---

### Decision · Program_Chairs · 2020-07-02

**Decision:**

Accept (Abstract only)

**Comment:**

The reviewers have identified that the authors have created a potentially useful dataset.

However, as a paper primarily about a dataset, the work lacks core details on the annotation guidelines in the paper description, and lacks quality assessment via measurement of agreement. The presented experiments themselves do not give an insight into the quality or value of the dataset beyond the specific scope of the QA data represented by the data set itself (i.e. there is no evidence provided that the dataset supports improvements in extrinsic search tasks, although this is suggested to be a motivation).

Overall, the authors have a potentially useful resource that requires more rigorous presentation. We would like to include an abbreviated (1 page) description of the resource, but encourage the authors to revise the work further.

We hope the feedback has been useful to you.

---

> ### Author Response · Authors · 2020-07-03
> **Thanks**
>
> Dear program chair,
>
> we appreciate your thoughtful decision and are happy to submit an abbreviated version of the paper.
> Will it be sufficient if we upload the new version before 6th of July?
>
> We also thank the reviewers for their efforts to dive into our paper, understand our approach as well as give detailed feedback as to what to improve.